# Pirimicarb Induction of Behavioral Disorders and of Neurological and Reproductive Toxicities in Male Rats: Euphoric and Preventive Effects of *Ephedra alata* Monjauzeana

**DOI:** 10.3390/ph16030402

**Published:** 2023-03-07

**Authors:** Latifa Khattabi, Aziez Chettoum, Houari Hemida, Walid Boussebaa, Maria Atanassova, Mohammed Messaoudi

**Affiliations:** 1Faculty of Nature and Life Sciences, University of Brothers Mentouri, Constantine1 (UFMC1), BP, 325 Route de Ain El Bey, Constantine 25017, Algeria; 2Biotechnology Research Center, Constantine (CRBt), Ali Mendjli Nouvelle Ville UV 03 BP E73, Constantine 25016, Algeria; 3Institute of Veterinary Sciences, University of Tiaret, Tiaret 14000, Algeria; 4Scientific and Technical Research Center in Physico-Chemical Analysis (CRAPC), BP384, Bou-Ismail, Tipaza 42004, Algeria; 5Nutritional Scientific Consulting, Chemical Engineering, University of Chemical Technology and Metalurgy, 1734 Sofia, Bulgaria; 6Nuclear Research Centre of Birine, Ain Oussera, Djelfa 17200, Algeria

**Keywords:** pirimicarb, toxicity, behavior, brain, reproduction, Il-1 β, histopathology, *Ephedra alata* monjauzena, fertility, UPLC-ESI-MS-MS/MRM

## Abstract

Carbamate pesticides are a risk to human well-being, and pirimicarb is the most widely employed carbamate insecticide. This ongoing investigation aimed to reveal its toxicity on neurobehavioral and reproductive function. The study was carried out on male Wistar rats by assessment of behavioral changes via experiments, such as the forced swim test and the elevated plus maze; determination of oxidative stress (checking parameters such as catalase activity, etc.); measurement of cortisol and testosterone serum titers, and IL-1β levels in the plasma and brain; and evaluation of histopathological lesions that induced pirimicarb after 28 days of gavage, specifically in the brain and testis. Traces of pirimicarb were analyzed in tissue extracts using LCMS/MS. At the same time, the beneficial and protective effect of EamCE (*Ephedra alata* monjauzeana Crude Extract) were tested. The outcomes showed considerable anxiety and depressive status, with an evident increase in cortisol and IL-1β titers and an important decrease in oxidative enzymes and testosterone. Significant histological lesions were also recorded. In addition, the LCMS/MS analysis affirmed the accumulation of pirimicarb in organ tissue from rats force-fed with pirimicarb. Conversely, EamCE demonstrated outstanding potential as a preventive treatment, restoring cognitive and physical performance, boosting fertility, enhancing antioxidant and anti-inflammatory activities and preserving tissue integrity. We concluded that pirimicarb has critical deleterious impacts on health, affecting the neuroimmune-endocrine axis, and EamCE has a general euphoric and preventive effect.

## 1. Introduction

Exposure to pesticides is expected for workers in their production and application; however, the public population can also be exposed from contaminated water and food. Pesticides have been extremely overused, and mostly uncontrolled in several developing countries. This may result in high exposure in large number of people and lead to more severe and widespread health effects [1]. Pesticide contamination comes from the circulation of the chemical through the target-treated part of the plants, resulting in environmental pollution. Such chemical residues impact human health via environmental and food contamination [2]. There are four common ways that pesticides can enter the human body: Dermal, oral, eye, and respiratory pathways. The toxicity of pesticides can vary depending on the type of exposure. As would be generally expected, the danger of pesticide contamination usually increases depending its concentration and its toxicity rate [3]. Highly hazardous pesticides are a risk to human well-being and the environment through enzymatic inhibition and the induction of oxidative stress [4]. Carbamate pesticides have been identified as disruptive agents of endocrine function, neuro-behavioral influencers, and having a role in increasing the risk of dementia [5]. Pirimicarb is a widely employed carbamate insecticide in apple orchards and in general, is considered to be a selective aphidicide [6]. Despite photodegradation and microbial degradation of pirimicarb, it has been proven [7] that pirimicarb still persists in some environmental matrices as a residue, notably in food [8,9,10]. Therefore, for this compound, food and water contamination present as the main ways to affect humans; however a video imaging technique has revealed vast deposition of pirimicarb on the bodies of greenhouse chrysanthemum workers [11], which is considered as direct exposure to pesticide nuisance. Furthermore, mammalian metabolism of pirimicarb involves hydrolysis of its moiety with subsequent demethylation at the dimethylamino group, resulting in the following metabolites that were detected in the urine samples of seven farmers who applied pirimicarb to their crops: 2-dimethylamino-5, 6-dimethyl-4-hydroxypyrimidine (DDHP), 2-methylamino-5,6-dimethyl-4-hydroxypyrimidine (MDHP), and 2-amino-5,6-dimethyl-4-hydroxypyrimidine (ADHP) [12]. There are few studies published concerning human toxicity of pirimicarb. These include a comparative study which was carried out to estimate the toxicity of nine pesticides, including pirimicarb, on three human cell lines (HepG2, HEK293, and JEG3) [13]. In addition, human volunteer investigations have been performed involving oral co-administration of pirimicarb (0.02 mg/kg/day) and chlorpyrifos-methyl at an acceptable daily intake, which is a very low (tolerable) concentration. Effectively, the results indicated no significant toxicokinetic interactions occurring between pirimicarb and chlorpyrifos-methyl when they were administrated together [14]. Otherwise, the majority of scientific surveys that were previously carried out on rodents and other laboratory animal models concerning the toxicity of pirimicarb only involved the compound in combination with other pesticide molecules, and not pirimicarb alone [15,16,17,18]. Due to the lack of data and the need for a better understanding of the toxicological effects of pirimicarb, we agreed the purpose of our study should be to elucidate the deleterious effects of pirimicarb on the brain, behavior, and reproductive system of male albino rats. We deliberately chose to investigate the preventive effects of *Ephedra alata* monjauzeana, as we were aware from our previous investigations [19] that this plant has many beneficial biological properties, namely antioxidant, anti-inflammatory, and others. More importantly, we consider this plant as a candidate for preventive effect due its ephedrine content [20,21]. Ephedrine has known neuroprotective effects against neuro-depression, ischemia injury and neurotoxicity [22,23,24,25]. In Algeria, *Ephedra alata* is used traditionally as a herbal tea to reduce stress, to enhance mood and to attenuate insomnia. 

## 2. Results

### 2.1. Behavioral Evaluation

#### 2.1.1. Forced Swim Test (FST)

In the FST, animals in G3 spent the majority of their time immobile (more than 69% of time) compared with control groups (G1 and G2) and G4 (treated with EamCE), in which a moderate amount of time was spent in swimming. Only 24.167 s was spent in escalation attempts; however, animals in the healthy groups spent the most time trying to climb (236.833 s (G1) and 226.500 s (G2)), as shown in Figure 1.

#### 2.1.2. Open Field Test (OFT)

Statistical analysis of OFT parameters showed that there were highly significant variations (*p* < 0.001). The total distance traveled by animals in the healthy control, EamCE control, and pirimicarb + EamCE groups was higher compared with the pesticide group (501.167 ± 3.656, 460.833 ± 3.817, 204.500 ± 3.271) vs. (94.000 ± 4.382) (Figure 2). Conversely, locomotor activity was significantly reduced in the group treated with pirimicarb, compared with the other control groups, and animals treated with the pirimicarb + EamCE complex.

#### 2.1.3. Elevated Plus Maze (EPM)

The locomotor activity of healthy controls animals (G1 and G2) was higher than that of animals treated with pirimicarb; this was evidenced by the total number of visits (11.167 ± 0.983 vs. 3.333 ± 0.816), (7.500 ± 1.049 vs. 3.333 ± 0.816). The same result was noted in the animals treated with pirimicarb + EamCE, which showed high locomotor activity compared with the animals treated with pirimicarb only. (11.667 ± 0.816 vs. 5.167 ± 1.472). Finally, we noted that the healthy control and EamCE-treated animals, and the animals treated pirimicarb + EamCE complex, spent the majority of time in the open arms of the device; this was well illustrated by the percentage of the time spent in the open fields (OF) of the device, as shown in Figure 3 (83.665 ± 0.731, 90.757 ± 1.244, 90.757 ± 1.244). At the same time, animals in the pirimicarb group spent the majority of their time (a percentage of 98.28 (±0.509)) in the closed fields (CF) of the device and a percentage of only 1.720 (±0.491) in the open parts (Figure 3).

#### 2.1.4. Elevated Zero Maze (EZM)

The latency of entering the open section for animals in the healthy control groups (G1 and G2) was lower than that of animals treated with pirimicarb (51.50 ± 1. 36 vs. 256.167 ± 0.983); (35.16 ± 1.47 vs. 102.333 ± 4.32). Conversely, the time spent in the open section was very high for healthy controls compared with animals treated with pirimicarb. In addition, the number of entries into the open section (EOS) and the number of head dips (HD) were most notable in the healthy groups, G1 (EOS): 8.16 ± 0.4, G2 (EOS): 5.5 ± 0.83, G1 (HD): 31.83 ± 1.32, G 22 (HD): 45.16± 0.40 and moderate in G4 animals treated with (pirimicarb + EamCE) complex, EOS: 3.5 ± 0.83, HD: 19.16± 1.32. But, both parameters were limited in the pirimicarb group, EOS (1.66 ± 0.81), HD (3.5 ± 0.83) (Figure 4).

### 2.2. Cortisol and Testosterone Titers

The serum dosage of cortisol showed a noticeable elevation in cortisol titer for the pirimicarb group compared with the other groups (G3: 2.070 ± 0.118 > G4: 1.353 ± 0.053 > G2: 0.977 ± 0.038 > G1: 0.996 ± 0.006), as illustrated in Figure 5.

In contrast, the testosterone titer (Figure 5) was lowest in the pirimicarb group and was at its highest level in the G2 animals treated with EamCE (G1: 4.908 ± 0.898, G2: 7.850 ± 0.388, G3: 0.809 ± 0.334, G4: 2.681 ± 0.166). *** *p* < 0.001 = very highly significant, and those with the same subscripts were not significantly different (*p* > 0.05).

### 2.3. Oxidative Stress Parameters 

Similarly, the tissue homogenates of the different organs studied had highly significant differences in the variation of protein concentrations (*p* < 0.001) (Table 1). MDA concentration was increased in G3, its value gradually decreasing from G4, to G2, to G1. Concentrations of GSH and catalase differed among the groups, being high in G1 and G2, moderately elevated in G4, and decreased in G3. The total findings are summarized in Figure 6.

### 2.4. IL-β Quantification

The quantification of IL-1β in brain homogenate and plasma samples showed a significant increase in animals in the G3 treated with pirimicarb (478.66 ± 1.43 pg/mL and 504.66 ± 0.22 pg/mL, respectively), compared with basic levels of IL-1β registered in samples from healthy animals (G1: 150 ± 0.68 pg/mL in brain homogenate and 155 ± 0.51 pg/mL in plasma). Concerning animals in G2 and G4, the levels of IL-1β were nearer to the basic values in G1 and were significantly reduced compared with the levels recorded in G4 (Figure 7).

### 2.5. Photomicrographs of Histologic Sections

Figure 8 demonstrates photomicrographs of the cerebral cortex of rats in different groups. Figure 9 shows photomicrographs of histological changes in the testis of animals in different groups.

### 2.6. UPLC-ESI-MS-MS Analysis of Brain and Testis 

The LCMS/MS analysis of extracts from brain and testis determined the presence of pirimicarb in all the tested extracts, the peaks for pirimicarb and its ion products (Table 2) are well represented in the chromatograms in Figure 10A–D corresponding to brain and testis extracts from rats in G3 and brain and testis extracts from rats in G4. Ion products structure is described in Figure 11. 

## 3. Discussion

Behavioral trials in rodents are utilized to estimate neurological traits and events, such as locomotor activity, depression-like behavior, socialization, memory, and other traits [26].The EPM and the OFT are some of the most widely employed procedures to investigate anxiety-like behavior in rats [27]. Moreover, the EZM has been proposed and validated as an apparatus for measuring anxiety status; however, the OFT has long been used for evaluating anxiety/fearfulness as well as locomotor/exploratory activity [28]. Likewise, the FST is also one of the most commonly used tests to assess depressive-like behavior in animal models [29]. The results obtained in the current study revealed the negative impact of pirimicarb on the behavior status of rats. In fact, in animals fed with pesticide (G3), the reduced total traveled distance, the minimized number of visits to the central area, and the increased duration of stay in the periphery zone and device corners indicated significantly decreased locomotor activity compared with control groups, and we noticed in G4 animals that the action of EamCE was clearly ameliorative. In addition, induced anxiety in the rats of G3 was significantly linked to reduced time spent in the open fields; increased time spent in the closed fields; fewer total number of entries into the open section; reduced frequency of head dipping; and extended latency to enter into the open section. Further, the climbing ability in FST was very feeble in affected rats of G3; they spent the majority of their time immobile. These deficiencies reveal clearly the diminution of the cognitive and locomotor competencies that would be sufficient to confirm the depressive effect of pirimicarb. Various negative effects on neuro-behavior have been previously recorded subsequent to exposure to a combination of different pesticides, even with safe doses (NOAEL) [30]. In the current study, this depressive effect was thoroughly reduced by the protective activity of EamCE in all the recorded behavioral parameters, with an obvious tendency to approach the values registered in the control groups. In point of fact, physical and cognitive performance in an anterior study was experimentally enhanced by the action of the major component ephedrine that was isolated from *Ephedra alata* plant, additionally to its many other potent pharmacological effects, notably for treating narcolepsy and depression [31]. 

Depressive and anxiety-like behaviors trigger the secretion of glucocorticoids [32,33]. Upon exposure to persistent or repetitive stress, this leads to already sensitive stress pathways become markedly hyperactive and, consequently, increases in cortisol secretion persist, which may cause alterations in glucocorticoid receptors and therefore contribute to the pathogenesis of mood and anxiety disorders [34]. Our findings appear to endorse the latter point because plasma titers of cortisol were high in anxious and depressive rats in the group treated with pirimicarb as a chemical stressor agent. The brain has long been considered a privileged organ, from an immunological point of view, since the blood–brain barrier and its tight junctions prevent the transmigration of systemic immune cells [35]. Cytokines are chemical messengers that stimulate the hypothalamus pituitary adrenal axis (HPA) axis when the organism is under stress or an infection, acting as endocrine factors to regulate hormone secretion and feedback control of the HPA axis. They transmit information from the immune compartments to the central nervous system as immunotransmitters and function in immunomodulatory neuroendocrine circuits [36]. Systemic immunological stressors elicit extended activation of the HPA axis, mainly due to the release of pro-inflammatory cytokines (IL-1, IL-6 and TNF-α) from stimulated peripheral immune cells [37]. There is ample evidence to support the association between increased cortisol and pro-inflammatory cytokines (IL-1β, IL-6, IL-8, TNF-α) in negative mood conditions, stress levels, anxiety, and depression [38]. IL1-β stimulates the production of mesencephalic astrocyte-derived neurotrophic factor (MANF), through its specific receptor type 1 (IL-1 R1) [39]. Astrocytes are usually described as maintenance cells that participate indirectly in nerve transmission. They are now known to participate in the inflammatory response in acute brain injury by modifying their morphological and functional phenotype; by expressing the major histocompatibility complex (MHC), cytokines, and chemokines; and by producing NO via over-expression of inducible NO synthase (iNOS) [40]. More than that, IL-1 aggravates brain damage and its pharmacological blockade or transgenic mutation of the IL-1 receptor reduces the size of the lesion and the behavioral dysfunction [41]. Given the above, we assume that the finding of high levels of IL-1β in both plasma and, notably, in brain homogenate of the pirimicarb-exposed animals had a direct and/or indirect action on the activation of HPA axis, the elevated titer of cortisol, the neurobehavioral disturbances and observed neurodegeneration. Vice versa, we strongly assert that cortisol enhanced the production of IL-1β from peripheral immune cells; this correlation has previously been proven [42]. Regarding the role of EamCE and the reduced release of IL-1β, we do associate this with the anti-inflammatory effect of the plant [19,43]. Importantly, *Ephedra alata* Decne has already shown the ability to reduce the production of pro-inflammatory cytokines [44].

The outcomes of oxidative stress assessment in our study illustrated the genesis of a substantial profile of oxidative stress, affecting both tissues of interest—the brain and testis. Indubitably, we refer to the samples from the pirimicarb-treated group of rats. Lipids are susceptible to oxidation, and lipid peroxidation products are potential biomarkers for oxidative stress status and its related diseases in vivo. Lipid peroxidation products in biological samples have been widely assessed; aldehydes such as MDA have been considered as a marker of lipid peroxidation in vivo [45]. Undoubtedly, the concentration of MDA as an oxidative stress indicator was increased in G3 rats exposed to pesticide, compared with the reduced concentration in G4, G2 and G1. A review work has reported that disturbances of MDA, GSH and SOD were associated with recurrent depressive disorder and lowered levels have been found in depressed patients compared with healthy volunteers [46]. Several investigations have been performed to evaluate the toxic effect of various types of pesticide and prove their ability to induce oxidative stress, as prominently indicated by low titers of antioxidant factors such as GSH, SOD, and catalase, and high concentrations of MDA [47,48] in brain tissue [49,50,51] and the male reproductive system [52,53,54], in the context of provoking neuro and reproductive male toxicities. Reactive oxygen species (ROS) are released as a result of normal cellular metabolism at low-to-moderate rates; they react normally in physiological cell processes, but higher amounts produce detrimental changes in cell elements, such as lipids, proteins, and DNA. The alteration of the oxidant/antioxidant balance in favor of oxidants is called “oxidative stress”. It may contribute to several health issues, including cancer, neurological disorders, hypertension, ischemia, diabetes, acute respiratory distress syndrome, etc. [55]. We suggest that pirimicarb exerted an inhibitory effect towards the antioxidant enzymes such as SOD and catalase, which may have resulted in an increase in ROS amounts (resulting even from pirimicarb metabolism) that in turn caused the elevation of MDA, the decrease in GSH, and the promotion of toxicological manifestations. A previous study noted changes in human red blood cell antioxidant enzymes in subjects with long-term exposure to pesticides. The most important finding was the reduction in SOD and catalase activities, with significant lower levels compared with controls in both the long and the short duration of pesticide exposure [56]. Nevertheless, the authors strongly presume that, EamCE enhanced the antioxidant potential of rats fed with the extract and protected them from damage caused by pirimicarb, notably the oxidative stress engendered in G4 rats. *Ephedra alata* has been widely exploited in different biological investigations. Unquestionably, the in vitro and in vivo antioxidant, anti-inflammatory, analgesic, antipyretic, antidiabetic, antihypolipidemic, antihemolytic, and antithrombotic activities of *Ephedra alata* extracts were efficiently performed, being rich in terms of polyphenol and flavonoid content [43,57,58,59,60,61,62]. Antioxidant phytochemicals, such as polyphenols and flavonoids, induce the high expression and activation of antioxidant enzymes, namely, catalase, SOD, glutathione peroxidase, and glutathione reductase. These plant components have electrophilic activity and can favor antioxidant enzymes via the Kelch-like ECH-associated protein 1-NF-E2-related factor-2 pathway and antioxidant responsive elements [63].

Histological examination of rats in control groups G1 and G2 showed brain sections with normal histological architecture with no lesions (Figure 8a,b). In the brain sections of pirimicarb-treated rats, the parenchymatous cells of the cerebral cortex showed degeneration and liquefactive necrosis, characterized by partial or complete dissolution of dead tissue and transformed into a liquid, viscous mass [64]. Prominent vacuolization was also identified; this was previously considered a side effect of the action of cytotoxic factors and its accumulation an important initiating event, causing metabolic alterations or stress responses that lead to cell death [65]. In addition, some glial cells showed pyknotic nuclei and numerous congested blood vessels (Figure 8d–f); these deleterious aspects have been determined in numerous neurotoxicity studies induced by pesticides or other toxic chemicals [66,67,68]. In contrast, the brain sections of G4 rats showed almost normal morphological appearance of nerve cells with less fine vacuolization. 

No pathological changes were observed in testicular sections of untreated control animals. Indeed, microscopic examination of testicular tissue of rats in the control group showed normal histological architecture with normal seminiferous tubules showing all cell layers of germinal cells and a spermatozoa-filled lumen (Figure 9, G1). However, EamCE (200 mg/kg) treated rats (G2) showed an increase in the number of spermatozoa and round spermatids (Figure 9, G2). Testes of rats treated with 147 mg/kg of pirimicarb (G3) showed marked histological changes characterized by reduced diameter; a markedly reduced number of spermatozoa; seminiferous tubules showing a disrupted germinal epithelium with various degree of atrophy; degeneration and necrosis with a prominent increase in interstitial space; and a reduced number of Leydig cells (Figure 9, G3). Rats receiving pirimicarb with EamCE (G4) showed a tendency towards a return to normal testicular histology indicated by a markedly developed germinal epithelium, and with almost a normal number of Sertoli cells and Leydig cells (Figure 9, G4). Moreover, the presence of spermatozoa was observed in the lumen of seminiferous tubules. The ongoing hispathological manifestations were accompanied by a raised level of cortisol and a decline mainly in the testosterone hormone. Previously, researchers have found that long-term exposure to deltamethrin (pesticide) caused alteration of reproductive hormones, including serious dysfunction of testicular tissue in which cortisol and testosterone levels were inversely proportional [69]. An earlier study that corroborates our findings on the potential role of carbamate pesticides interference on the natural hormones of the hypothalamic–pituitary–thyroid that result in disturbances of the male reproductive system. It reported that there is evidence that exposure to carbamates leads to lower levels of gonadotropin-releasing hormone (GnRH), luteinizing hormone (LH) and/or follicle-stimulating hormone (FSH), thus compromising steroid genesis and spermatogenesis. Furthermore, various histologic alterations have been demonstrated in the testis along with deficiency of male reproductive capacity [70].

The protective effect of EamCE regarding testicular tissue integrity and boosting fertility via enhancing spermatogenesis and increasing testosterone level, could probably be explained by the effect of saponins [71,72] and ephedrine alkaloids [73,74] that are contained in the plant crude extract. In effect, saponins have been known as substances responsible for, and enhancers of, an aphrodisiac effect due to their ability to increase androgen hormones [75,76]. In addition, we also observed, from a few days after starting the experiment, and frequently until the day of sacrifice, some sexual behaviors in the G2 (EamCE control) fed rats, such as mounting and intermission, that strongly explain the presumed fertilizing and aphrodisiac effects of EamCE. 

Contrary to organophosphate poisoning, carbamate poisoning normally starts to diminish within a few hours and disappears after 24 h, usually without any permanent sequelae. Carbamates commonly do not traverse the blood–brain barrier as easily as organophosphates; as such, brain injuries with carbamates occur with a lower frequency and are less severe than with organophosphates [72]. In another study, it was reported that central nervous system symptoms are not particularly noticeable in carbamate poisoning due to the poor permeability of these compounds across the blood–brain barrier [73]. A molecule can be totally (100%) absorbed from a given formulation; however, it may have low bioavailability before being broken down after absorption [74]. Pesticides are generally distributed in the organism due to their ability to bind with plasma proteins, blood cells, and lipids in various organs and peripheral tissues. The binding potential is determined by the lipophilicity, which increases the pesticide’s successive bioaccumulation. Thus, the lipophilicity of compounds can truly alter their bioavailability [75]. According to the Swiss ADME prediction online platform [76], pirimicarb has a lipophilicity with a log Po/w of 3.39 and with a score of bioavailability equal to 0.55, which are sufficient values for a molecule to accumulate and engender its different modes of action. LCMS/MS-MRM analysis in the current study has effectively allowed us to confirm simulation data and find pirimicarb in brain and testis tissues from animals orally administered pirimicarb (rats in G3 and G4). The revelation of pirimicarb in brain and testis disclose that this molecule is resistant to different mechanisms of biotransformation and metabolism in the rat organism; in this case, pirimicarb bioavailability increased and the molecule was able to infiltrate many types of tissue. Renal excretion of unchanged drug has only a modest role in the complete elimination of many chemicals as long as lipophilic compounds filtered through the glomerulus are mostly reabsorbed into the systemic circulation during passage through the renal tubules [77].

Finally, we have summarized the impact of pirimicarb in the current study, and the proposed mechanism of action of EamCE with candidate metabolites at each level of interaction is Figure 12.

## 4. Materials and Methods

### 4.1. Chemicals and Reagents

Pirimor 50 DG was obtained from Syngenta; sodium chloride (NaCl), Bradford reagent, bovine serum albumin (BSA), Trizma (tris), TCA (trichloroacetic acid), thiobarbituric acid (TBA), hydrogen chloride (HCl), 5,5-dithio-bis-(2-nitrobenzoic acid (DTNB), ethylenediaminetetraacetic acid (EDTA), pyrogallol, hydrogen peroxide (H_2_O_2_) 30%, monosodium phosphate (NaH_2_PO_4_,2H_2_O), disodium phosphate disodium phosphate (Na_2_HPO_4_,2H_2_O), potassium phosphate monobasic (KH_2_PO_4_), acetonitrile ≥ 99.8%, formic acid 98–100%, magnesium sulfate (MgSO_4_), and activated charcoal were all obtained from Sigma Aldrich; ethanol 96% xylene and formaldehyde 37–38% were obtained from PanReac AppliChem; Certistain was obtained from Merck, Mayer’ hematoxylin was obtained from Specilab, and neoxylene was obtained from Eukitt.

### 4.2. Animals

Twenty-four albino Wistar male rats were purchased from Algiers Pasteur Institute (IPA), weighting 190 g–230 g and underwent an acclimatization period of 15 days before beginning the experiment. The rats were kept at 22 ± 2 °C and 50–60% humidity under a light/dark cycle of 12 h and had free access to standard commercial pelleted feed (supplied by ‘‘ONAB” Guelma, Algeria) and clean tap water ad libitum at the animal laboratory of CRBt.

### 4.3. Experimental Design

In order to induce subacute toxicity, we followed different protocols [49,77,78,79] to constitute our own. The animals were divided in four groups (6 rats/each) that received different treatments by oral gavage: G1, deionized water; G2, 200 mg/kg of EamCE [19]; G3, 14.5 mg/kg (1/10 of LD 50 (145 mg/kg)) of pirimicarb [12,15,80,81]; and G4, 14.5 mg/kg (1/10 of LD 50 (145 mg/kg)) of pirimicarb + 200 mg/kg of EamCE (the EamCE was administrated 1h before administrating pirimicarb). These daily repeated doses were given for a period of 28 days. Thereafter, the animals were subject to three successive days of behavioral examination (Figure 13). Subsequently, animals were euthanized by cervical dislocation; blood tissue and some organs (brain, liver, spleen, kidney and testicles) were collected for carrying out further investigations. All procedures were accomplished in accordance with laboratory guidelines for animal care, Algerian Executive Directive (18 March 2004, N◦ 10–90 JORA) and Law No.88-08 of 26 January 1988 relating to veterinary medicine activities and the protection of animal health (N◦ JORA: 004 of 27-01-198).

### 4.4. Behavioral Evaluation

#### 4.4.1. Forced Swim Test (FST)

The FST in rats is a preclinical behavioral model that has good predictive validity and is widely used to determine the efficacy of antidepressant drugs (ADS) [82]. For the current study, our purpose was to check the eventual depressant effect of pirimicarb. Therefore, we executed the FST [83] with some modifications. We eliminated the first step (pre-test) that consisted of inducing a stressful situation in rats, because the rats were supposed to be already depressed under the effect of pirimicarb. Therefore, the animals were directly immersed in an aquarium (54 cm high by (34 × 60 cm) base area; these dimensions ensure that the rat cannot escape by clinging to the edges of the device) for five minutes. The behavior of the animal in the device was filmed using a video camera. The aquarium was filled with warm water (26 °C) up to a height of 40 cm, in order to ensure that the rat would not use its lower limbs to stay on the surface and therefore be forced to swim. We then proceeded to analyze the sequences and measure the time spent in immobility, swimming and climbing (escalation).

#### 4.4.2. Open Field Test (OFT)

The locomotor activity of rats was measured using a device consisting of a rectangular wooden enclosure 1 m in diameter and 50 cm high, divided into 7 parts each of the same area: a central part and six peripheral parts. The central part serves as a starting point for the animals in each test. The animals are placed in the device for 10 min. Locomotion in the OFT was evaluated by noting the total distance traveled, the number of entries in the central part and the number of redress. These cumulative indices gave us the total locomotion index for rats in the device [84].

#### 4.4.3. Elevated Plus Maze (EPM)

The EPM device was in a form of a cross raised to a height of 40–60 cm from the ground. It consisted of a central part (10 × 10 cm), and two open protected arms without walls (50 × 10 × 50 cm) which oppose two other arms with closed walls, which are perpendicular to the open protected arms. The test lasts 5 min and begins when the rat is placed in the center of the maze, facing an open arm. An animal that explored within the open arms was described as being “slightly anxious” and an animal that remained confined in the closed arms of the device was described as being “anxious”. Two types of variables were identified: classic variables [85,86] and more ethological variables taken from the defensive behavioral repertoire of rodents [87].

#### 4.4.4. Elevated Zero Maze (EZM)

The EZM was an elevated annular runway with alternating open and enclosed quadrants (105 cm diameter, 10 cm width) 65 cm above from the ground level, divided equally into four areas; this updated device helped to remove any uncertainty of interpretation regarding the time lost on the central square of the traditional design and allowed uninterrupted exploration [88]. Latency to enter into an open section, time spent in the open sections, number of entries in the open sections and number of head-dips were measured for 5 min [28].

### 4.5. Cortisol and Testosterone Titers

Serum titers of cortisol and testosterone were quantified using Abbott Alinity automaton with their respective specific kits (Alinity 08P3320, Alinity 07P6821).

### 4.6. Oxidative Stress Parameters

#### 4.6.1. Tissue Homogenate

The organs were immediately collected, washed using 0.9% NaCl solution and weighed; 1 g of each organ was put in 2 mL of TBS (Tris-buffered saline): Tris 50 mM, NaCl 150 mM, adjusted to pH = 7.4 with HCl 1 M. The mix was homogenized using a “SONICS, Vibra-Cell VX 130” sonificator, under ice-cold conditions. Homogenates were centrifuged at 3000× *g* for 30 min at 4 °C. The supernatants were then aliquoted and stored at −20 °C.

#### 4.6.2. Protein Titration

Proteins from tissue homogenates were quantified spectrophotometrically at 595 nm according to the modified method of Bradford [89], using bovine serum albumin as standard.

#### 4.6.3. Malondialdehyde (MDA)

The evaluation of lipid peroxidation levels was accomplished by detecting the value of MDA in organ homogenates. MDA reacts with thiobarbituric acid as a reactive substance to generate a red-colored complex. The procedure involved combining 500 µL of tissue homogenate with 1 mL of TCA-TBA-HCI (15%, 0.375%, 0.25 N) and mixing thoroughly. The mixture was heated in a boiling water bath for 15 min. Then, the flocculent precipitate was removed by centrifuging at 1000× *g* for 10 min and the absorbance was measured at 535 nm [90].

#### 4.6.4. Reduced Glutathione (GSH)

GSH levels of organ homogenates were measured by employing a colorimetric technique based on the oxidation of GSH by DTNB, which generates a yellow color according to the Elman method [91]. Tissue homogenate (400 µL) was added to 100 uL of sulfosalicylic acid (0.25%) and left for 15 min in an ice bath. After centrifugation at 1000 rpm for 15 min, 250 µL of supernatant was collected and added to 500 µL mL of Tris-EDTA buffer (0.4 M HCl, 0.02 M EDTA, pH 9.6) and 12.5 µL of DTNB (0.01 M). After shaking and incubation for 5 min, the absorbance was recorded at 412 nm.

#### 4.6.5. Superoxide Dismutase (SOD)

The SOD activity was estimated according to the Marklund and Nandy procedures [92,93] with slight changes. The method was based on inhibition of the auto-oxidation of pyrogallol by SOD. An 850 µL quantity of Tris HCL buffer (50 mM, pH = 8.2) was added, followed by 100 µL of EDTA (10 mM). Then, the reaction was started by the addition of 50 µL of pyrogallol (2.5 mM in 10 mM HCL). The absorbance reading was taken at 420 nm every minute for 3 min in the presence or absence of 20 µL of tissue homogenate sample. SOD activity was expressed as U/mg protein. One unit of SOD activity (U) was determined as the amount of enzyme required to inhibit 50% of pyrogallol autoxidation.

#### 4.6.6. Catalase (CAT)

CAT activity was evaluated following the procedure of Aebi [94]. In brief, 983.5 μL of H_2_O_2_ (10 mM, prepared in 50 mM phosphate buffer (KH_2_PO_4_, Na_2_HPO_4_), pH = 7.2) was added to 16.5 μL of tissue homogenate. The reaction was based on the disappearance of hydrogen peroxide, and the decrease in absorbance was monitored for 30 s at 240 nm.

### 4.7. IL1-β Titration and Quantification in Brain and Plasma

The level of Il-1β was quantified from brain homogenates and plasma samples, using Rat IL-1β ELISA Kit, E-EL-R0012 (Elabscience Biotechnology Inc.: Houston, TX, USA). The generation of a standard curve and quantification steps were executed according to the manufacturer’s handbook.

### 4.8. Histopathological Examination

No treatment-related deaths were evident. Sacrificed rats were subjected to a full necropsy examination. Organs were then removed and examined for any gross lesion after being rinsed with NaCl (0.9%) solution thoroughly and properly. Then, they were immediately fixed in formaldehyde solution (10%). Tissue samples were routinely processed through an automatic tissue processor. After that, the tissues were embedded in paraffin, sectioned, and stained with hematoxylin and eosin (H&E) according to the technique described by [95]. Photomicrographs of selected lesions were taken using an optical microscope with an integrated camera (BioBlue Euromex (EU 2131898)) and treated by Image Focus plus V2.

### 4.9. Testing for the Presence of Pirimicarb in Brain and Testis Tissues

#### 4.9.1. Pirimicarb Extraction

A 0.33 g tissue fragment from each animal target organ (brain and testis) was mixed and stirred vigorously with 10 mL of distilled water, 10 mL of acetonitrile, 4 G of MgSO_4_ and 4 G of NaCl. After that, it was centrifuged at 4500 rpm, 15 °C for 5 min. The supernatants were collected, and 2 g of MgSO_4_ and 25 mg of activated charcoal were added. The newly constituted mixture was centrifuged under the same previous conditions, and the supernatants were recovered, filtered through a 0.22 µM filter and evaporated. Next, the extract was dissolved with a small quantity of acetonitrile and stored at 20 °C before analysis [96].

#### 4.9.2. UPLC-ESI-MS-MS Analysis

The tissue extracts were analyzed using the LC-MS/MS method in multiple reaction monitoring (MRM) mode. The analysis was performed using UPLC-ESI-MS-MS Shimadzu 8040 Ultra-High sensitivity with UFMS technology and equipped with binary bump Nexera XR LC-20AD. The ESI conditions were as follows: CID gas, 230 KPa; conversion dynode, −6.00 Kv; interface temperature, 350 °C; DL temperature, 250 °C; nebulizing gas flow, 3.00 L/min; heat block, 400 °C; and drying gas flow, 15.00 L/min. The MRM transition was accessed from Shimadzu Pesticide MRM Library Support for LC/MS/MS. The pump mode was isocratic, and the mobile phase contained: 15% A water, 0.1% formic acid, and 85% B Acetonitril. The flow rate was: 0.2 mL/min and the injected volume of extracts was 5 µL, using a Restek column of force C18 1.8 µm 50 × 2.1 mm.

### 4.10. Statistical Study

The outcomes of our experiments were expressed as means and standard deviations related to six values for each group and each studied parameter. Variance analysis was conducted on XLSTAT Version 2016.02.28451 using ANOVA, the significance of differences was checked using Tukey’s HSD test. Values with different subscripts (a, b, c, d, e) in the same parameter were significantly different compared with the healthy group G1 (*** *p* < 0.001 = very highly significant, ** *p* < 0.01 = highly significant, * *p* < 0.05 = significant), and those with the same subscripts were not significantly different (*p* > 0.05).

## 5. Conclusions

Pesticides are recognized as producing toxicological effects with various modes of action. Indeed, the outcomes of the prevailing research gave us sufficient insight about pirimicarb toxicity on neurobehavior and reproductive system. The proven accumulation of pirimicarb in the brain and testis explained perturbations of behavior and mood, induction of oxidative stress, and inflammation status with serious lesions in brain tissue and testis. These manifestations are characteristic of traumatic disorders such as dementia and infertility. In the long term, they may lead to more severe consequences, such as cancer malignancies and death. EamCE from *Ephedra alata* monjauzeana offered significant protection from pirimicarb damage due to its antioxidant, anti-inflammatory, fertilizing and euphoric potential. In future projects, we plan to track pirimicarb circulation with precise amounts and to elucidate its mode of action in several organs in male and female rats, in conjunction with investigating whether either EamCE or specific individually separated molecule/molecules from *Ephedra alata* monjauzeana is/are more efficient.

## Figures and Tables

**Figure 1 pharmaceuticals-16-00402-f001:**
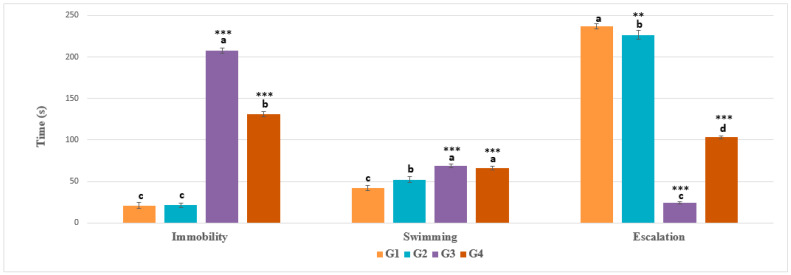
**Evaluation of FST parameters**. G1: Negative control, G2: Negative control of EamCE, G3: EamCE + pirimicarb, G4: Pirimicarb. The FST parameters included immobility, swimming, and escalation times; the histogram in Figure 1 describes their variation between different experimental groups. Time was recorded in seconds (s). *** *p* < 0.001 = very highly significant, ** *p* < 0.01 = highly significant, and those with the same subscripts were not significantly different (*p* > 0.05).

**Figure 2 pharmaceuticals-16-00402-f002:**
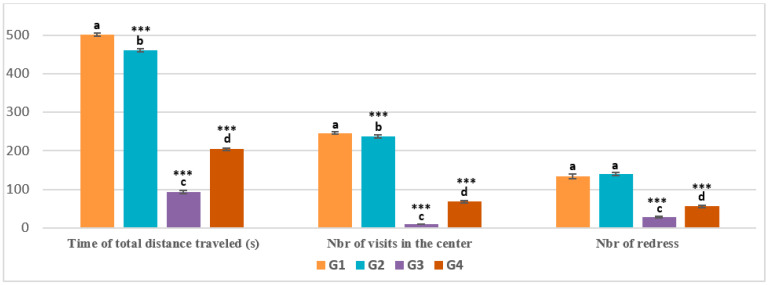
**Evaluation of OFT parameters.** Animals in the OFT were evaluated by measuring the duration of total distance traveled, the number (nbr) of visits to the center of the device, and the nbr of redress. Time was recorded in seconds (s). *** *p* < 0.001 = very highly significant, and those with the same subscripts were not significantly different (*p* > 0.05).

**Figure 3 pharmaceuticals-16-00402-f003:**
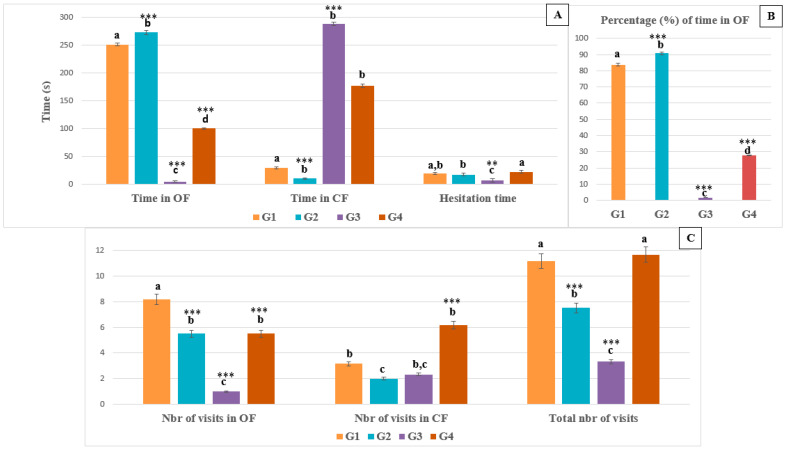
**Evaluation of EPM parameters**. For the EPM, several parameters were taken in consideration, including time spent in open fields (OF) and in closed fields (CF), hesitation time (A). Percentage of time in OF (B). The number of visits in different compartments of the apparatus (C). Time was recorded in seconds (s). *** *p* < 0.001 = very highly significant, ** *p* < 0.01 = highly significant, and those with the same subscripts were not significantly different (*p* > 0.05).

**Figure 4 pharmaceuticals-16-00402-f004:**
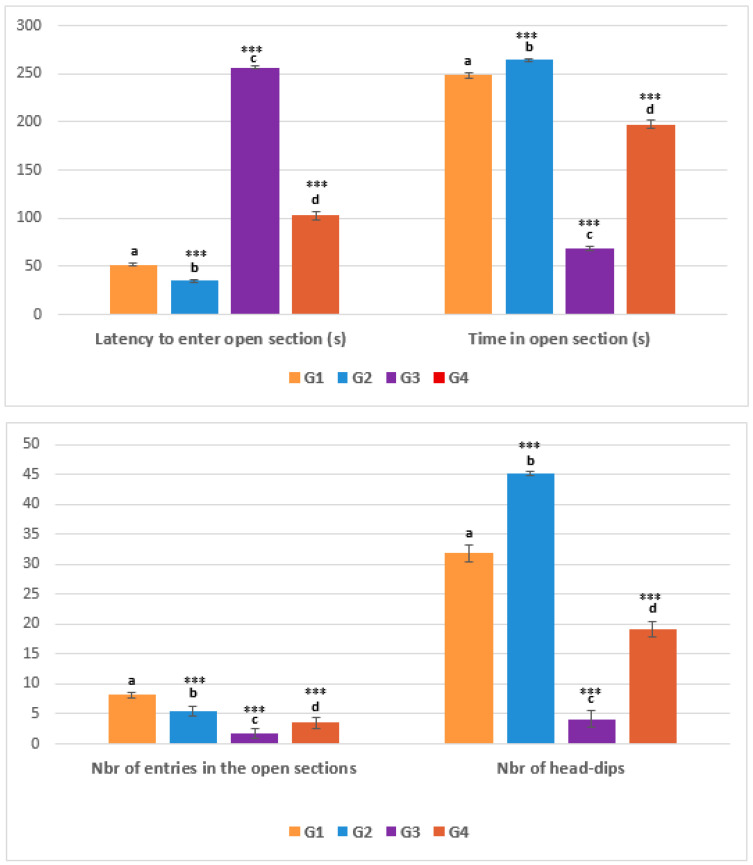
**Evaluation of EZM parameters.** In this trial, we recorded the latency time and the duration spent in the open section of the device in seconds (s); we also assessed the number of entries in the open section and the number of head-dips. *** *p* < 0.001 = very highly significant, and those with the same subscripts were not significantly different (*p* > 0.05).

**Figure 5 pharmaceuticals-16-00402-f005:**
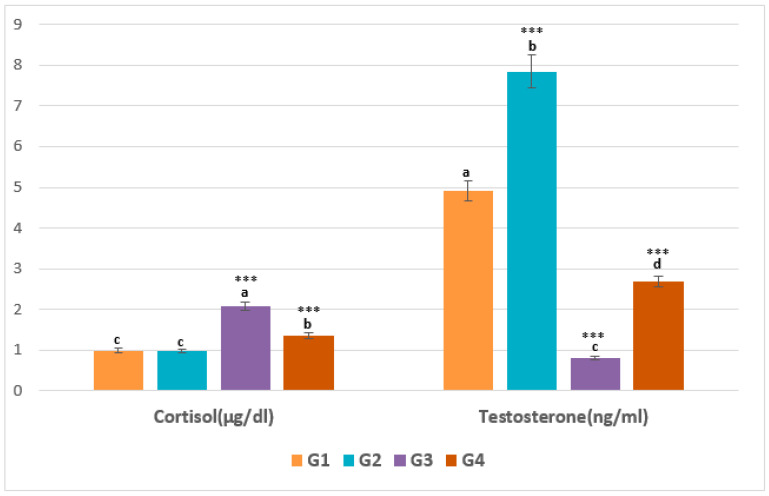
Serum titers of cortisol and testosterone.

**Figure 6 pharmaceuticals-16-00402-f006:**
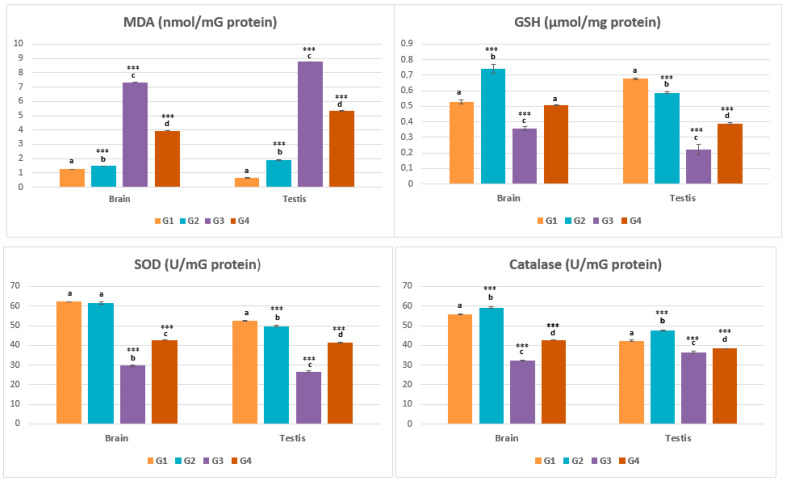
**Assessment of oxidative stress biomarkers.** The figure demonstrates the concentration of: malondialdehyde (MDA), reduced glutathione (GSH), superoxide dismutase (SOD) and catalase in tissue homogenates of brain and testis. *** *p* < 0.001 = very highly significant, and those with the same subscripts were not significantly different (*p* > 0.05).

**Figure 7 pharmaceuticals-16-00402-f007:**
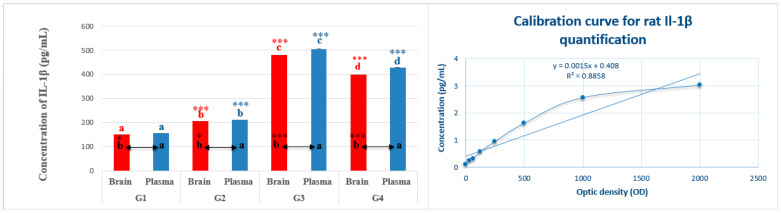
**Quantification and titration of IL-1β in brain and plasma**. In Figure 7, the calibration standard curve (concentration of IL-1β (pg/mL) vs. OD) is represented on the right. The red subscripts and asterisks indicate statistical comparisons of significance between IL-1β levels in brain homogenate of different groups (G1, G2, G3, G4). The blue subscripts and asterisks indicate statistical comparisons of significance between IL-β levels in plasma of the different groups. Black subscripts and asterisks indicate statistical comparisons between IL-1β levels in the brain homogenate and plasma of each group. *** *p* < 0.001 = very highly significant, * *p* < 0.05 = significant, and those with the same subscripts were not significantly different (*p* > 0.05).

**Figure 8 pharmaceuticals-16-00402-f008:**
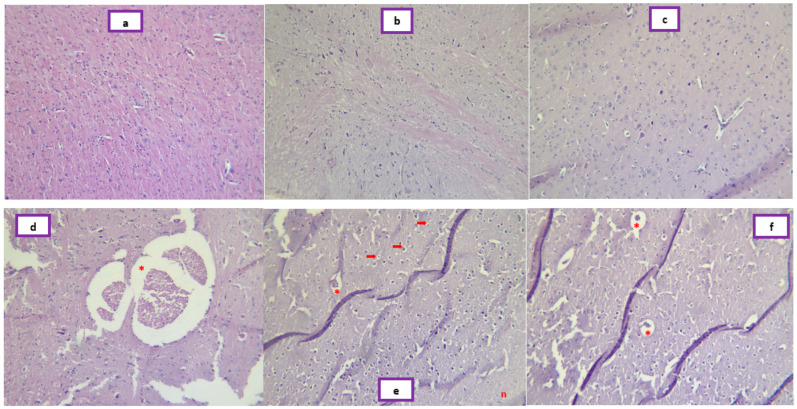
**Photomicrographs of cerebral cortex sections**. Images (**a**,**b**) are control groups G1 and G2 showing normal neuroglial cell arrangement. G3 is represented in images (**d**,**e**,**f**) showing an area of liquefactive necrosis (n), shrunken nerve cells with dark nuclei and surrounded by vacuoles (red arrows), vascular congestions (asterisks) and vacuolization. Image (**c**) shows a G4 section with almost normal morphological appearance of nerve cells with less fine vacuolization (H&E, 10×).

**Figure 9 pharmaceuticals-16-00402-f009:**
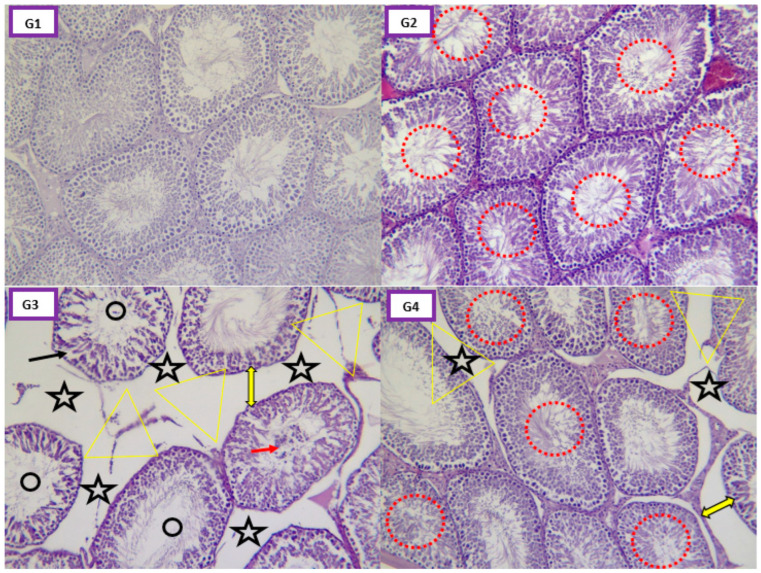
**Photomicrographs of testis sections**. Control group rats (**G1**) showed typical organization of seminiferous tubules. EamCE (200 mg/kg) treated rats (**G2**) showed important spermatogenesis (red circle), normal testicular architecture and normal Leydig cell appearance. Rats treated with 147 mg/kg of pirimicarb (**G3**) exhibited degeneration of germinal and Sertoli cells (black arrow); absence of Leydig cells (black star); loss of interstitial tissue (yellow triangle); an abnormal increase in intertubular spaces (yellow double arrow); spermatocytic maturation arrest (black circle); lumen-devoid spermatozoa (black star) and the formation of many exfoliated cells (red arrow). Rats treated with 147 mg/kg of pirimicarb and EamCE (200 mg/kg) (**G4**) showing an increase in spermatogenesis (red circle) and an improvement in general appearance almost comparable to the control group, except for partial loss of Leydig cells (black star) and interstitial tissue (yellow triangle) (H&E, 10×).

**Figure 10 pharmaceuticals-16-00402-f010:**
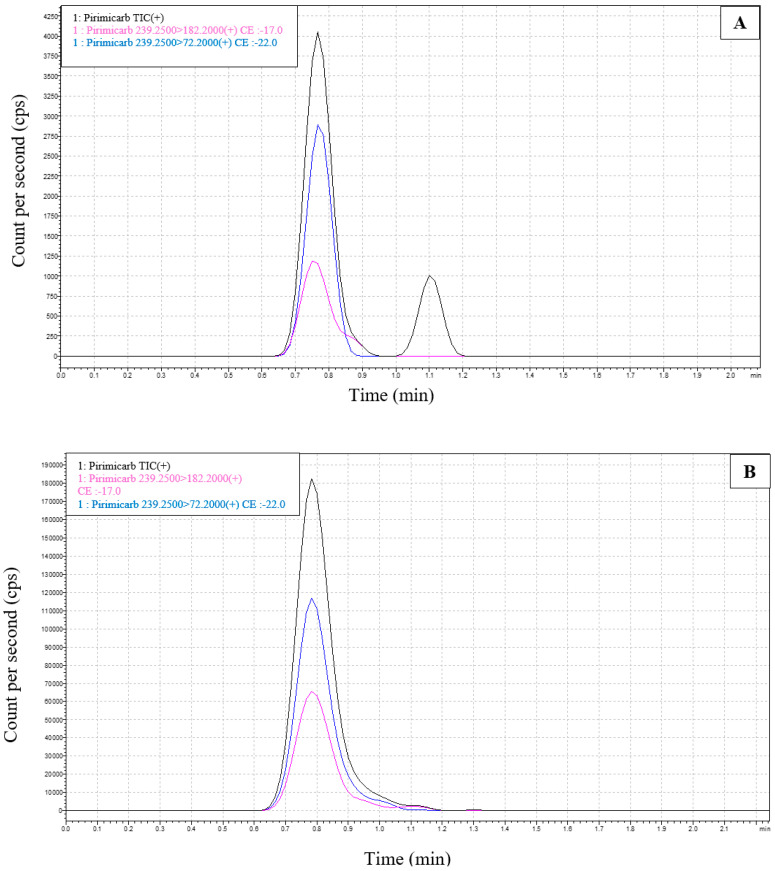
**Representative chromatograms of pirimicarb**. (Rt 0.8 min) from extracted samples. (**A**) Chromatogram of pirimicarb from brain extracts of G3 rats; (**B**) chromatogram of pirimicarb from testis extracts of G3 rats; (**C**) chromatogram of pirimicarb from brain extracts of G4 rats; (**D**) chromatogram of pirimicarb from testis extracts of G4 rats.

**Figure 11 pharmaceuticals-16-00402-f011:**
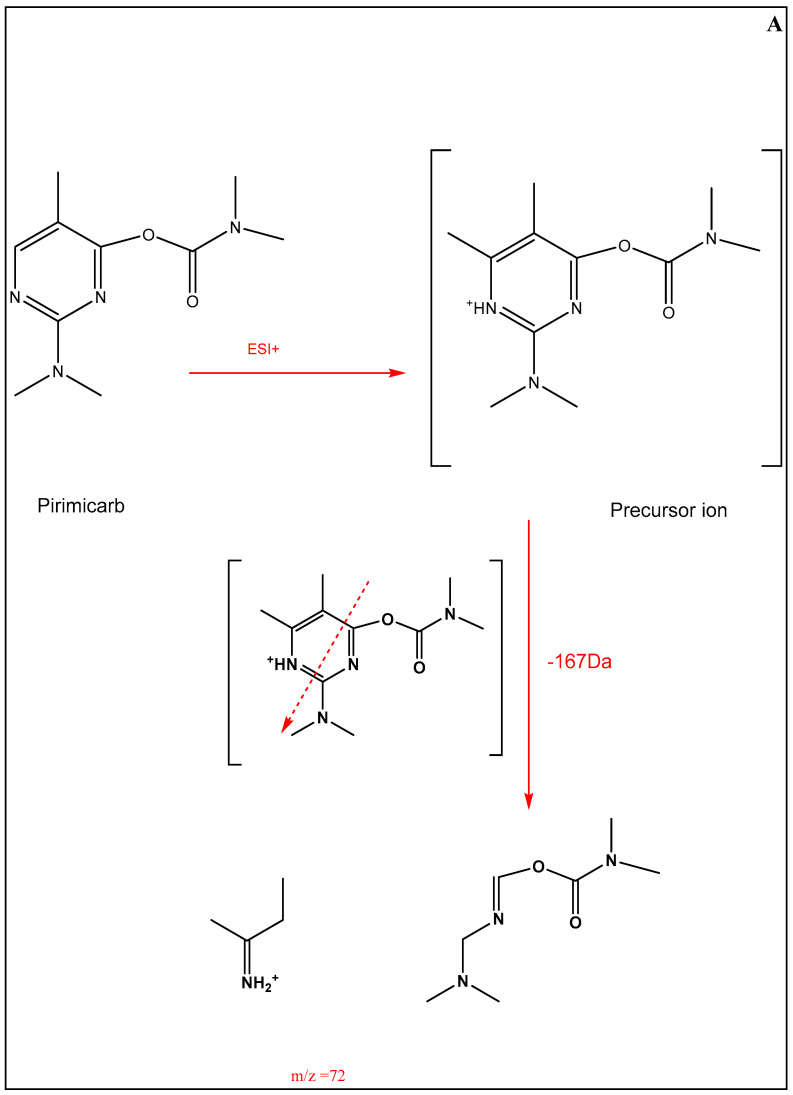
The chemical structure of 72 and 182 ion fragments. (**A**): The chemical structure of 72 fragment, (**B**): The chemical structure of 182 fragment.

**Figure 12 pharmaceuticals-16-00402-f012:**
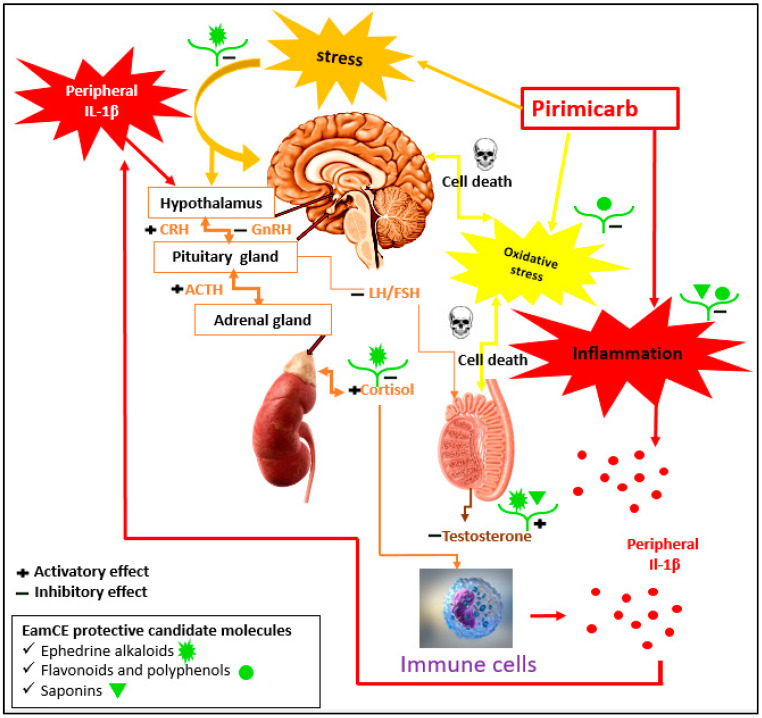
**Proposed mechanisms elucidating how EamCE protects against pirimicarb-induced damage.** Pirimicarb induces oxidative stress in the brain and testis (provoking cell death and serious lesions), causing inflammatory status by the release of IL-1β from activated immune cells. In addition to its role in the induction of stress; that is responsible for behavioral changes and activation of the HPA via stimulating the hypothalamus to secrete corticotropin-releasing hormone (CRH) and down regulating GnRH. CRH activates the pituitary gland to release adrenocorticotropic hormone (ACTH) and decreases levels of GnRH; this downregulates the release of LH and FSH from the pituitary gland. The result is a low amount of testosterone and a high secretion of cortisol. Cortisol reaches the immune cells and activates them to secrete more IL-1β; increased peripheral IL-1β concentrations reach the brain and activate secretion of cortisol via specific receptors. EamCE produces a general protective effect due its metabolites, as illustrated in Figure 12.

**Figure 13 pharmaceuticals-16-00402-f013:**
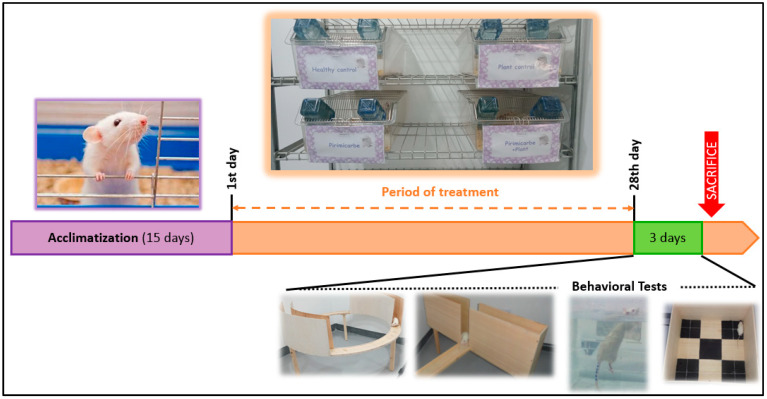
**Experimental design. Figure 13** summarizes the timeline of the principal steps that were undertaken in our in-vivo experiment: the acclimatization period before starting the experiment, allowing rats to adapt to the new animal house environment; the period of treatment, in which daily force-feeding of the different groups of rats with adequate products and doses took place; and behavioral tests, in which rats from all four groups underwent behavioral testing procedures (FST, OFT, EPM, EZM).

**Table 1 pharmaceuticals-16-00402-t001:** Protein concentration estimations in the brain and testis. *** *p* < 0.001 = very highly significant, and those with the same subscripts were not significantly different (*p* > 0.05).

	Tissues	G1	G2	G3	G4
Protein concentration (µg/mL)	Brain	(2.372 ± 0.001) ^a^	(1.714 ± 0.003) ^b***^	(1.983 ± 0.003) ^c***^	(1.829 ± 0.002) ^d***^
	Testis	(1.844 ± 0.003) ^a^	(2.053 ± 0.002) ^b***^	(2.254 ± 0.002) ^c***^	(2.042 ± 0.002) ^d***^

**Table 2 pharmaceuticals-16-00402-t002:** LCMS/MS data of pirimicarb.

Compound	Formula	M	MH^+^	Ionization Mode	MRM Transition	Collision Energy (v)	Rt Min
Pirimicarb	C_11_H_18_NO_2_	238.1430	239.15	ESI+	72.1	−22	0.8
					182.2	−17	0.8

## Data Availability

Data is contained in the article.

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
