# Peer review of "Pirimicarb Induction of Behavioral Disorders and of Neurological and Reproductive Toxicities in Male Rats: Euphoric and Preventive Effects of Ephedra alata Monjauzeana"

_pharmaceuticals, 2023, doi:10.3390/ph16030402_

Round 1
Reviewer 1 Report
In this research, the authors have investigated the toxicity of Pirimicarb on the neurobehavioral and reproductive functions of male Wistar rats, as well as the preventive role of Ephedra alata monjauzeanaare in this field. The authors have presented an interesting work, however, the present manuscript can be considered for publication after major revision. The detailed comments are listed as follows.
- Many words are stuck together in the text, which needs to be revised. For instance, ‘Carbamatepesticides’ in line 22, ‘preventivepotentials’ in line 32, ‘frombrain’ in line 98, ‘FSTis’ in line 215, ‘experimentswere in line 455, ….
- “PicimicarbInduction” in the title should be replaced by “Pirimicarb Induction”
- Please define in the text the meaning of each acronym or abbreviation before starting to use it. For example: “EamCE” in line 27.
- In line 338 authors stated that for G3: 1/10 LD 50/kg of pirimicarb was used. However, they didn’t present the amount of LD50. Besides, information about the LD50 and SHD of the pirimicarb toxicant is important in the toxicology discussion.
- In line 338 “of pirimicarb[53], [15], [54], [12] of pirimicarb, ..” of pirimicarb is repeated.
- Provide the reason for choosing the Wistar rat instead of other species such as CD-1, ICR-Swiss or Sprague Dawley in this research.
- In line 85 ‘FST test’ should be replaced by ‘FST’ or ‘FS test’.
- All figures 1 to 6 are repetitions of tables 1 to 6. Authors should present results either as tables or figures. It is recommended to use histogram figures and delete the corresponding tables.
- “4. Photomicrographs of histologic sections’ in line 167 should be corrected.
- In Figure 7, the subfigures corresponding to G3 treatment is presented 3 times. please explain the reason. Is the magnification different? The difference between the shapes should be marked by (a), (b), (c), (d), (e) and (f).
- Figure 9 is blurry.
- Title of Table 7 “Table 7: LCMS/MS data of pirimicarb” should be presented above the table.
- In line 208, “2. Discussion” should be changed to “3. Discussion”
- The manuscript suffers from supportive materials and discussion. A comparative study on the previous research findings needs to be included.
- In line 348, “Figure 6: experimental design” the number of figure should be corrected.
- Although statistical analysis appears to have been performed, there is no text in the methods section providing the details about the tests of the assumptions of ANOVA (normality and homogeneity of variance).
Author Response
Dear referee;
We are grateful for your valuable time and useful contribution. We do appreciate the inputs that you have given and we believe that it will definitely help to improve our manuscript. You would find in the paper all the required changes/ additions highlighted in green. Also, you would find below responses for each of your requirements.
- Many words are stuck together in the text, which needs to be revised. For instance, ‘Carbamate pesticides’ in line 22, ‘preventive potentials’ in line 32, ‘from brain’ in line 98, ‘FST is’ in line 215, ‘experiments were in line 455, ….
Response:
- Every two attached words were separated and highlighted in yellow
- Please define in the text the meaning of each acronym or abbreviation before starting to use it. For example: “EamCE” in line 27.
Response:
- EamCE was defined in its first mention and highlighted in green
- In line 338 authors stated that for G3: 1/10 LD 50/kg of pirimicarb was used. However, they didn’t present the amount of LD50. Besides, information about the LD50 and SHD of the pirimicarb toxicant is important in the toxicology discussion
Response:
- The LD50 of pirimicarb was added and discussed
- In line 338 “of pirimicarb [53], [15], [54], [12] of pirimicarb, ..” of pirimicarb is repeated.
Response:
- The repetition was omitted
- Provide the reason for choosing the Wistar rat instead of other species such as CD-1, ICR-Swiss or Sprague Dawley in this research.
Response:
- We have used rats instead of mice in order to collect a sufficient quantity of blood and sufficient mass of tissues that are necessary for several investigations.
- The Wistar species is the unique available line of rats in Algeria Pasteur Institute
- The Wistar rat is largely used for medical and biological research experiments. Notably, it is used as a standardized species for toxicological studies.
- In line 85 ‘FST test’ should be replaced by ‘FST’ or ‘FS test’.
Response:
- The word of test was omitted
- All figures 1 to 6 are repetitions of tables 1 to 6. Authors should present results either as tables or figures. It is recommended to use histogram figures and delete the corresponding tables.
Response:
- The tables were replaced by figures
- “ Photomicrographs of histologic sections’ in line 167 should be corrected.
Response:
- Corrected
- In Figure 7, the subfigures corresponding to G3 treatment is presented 3 times. Please explain the reason. Is the magnification different? The difference between the shapes should be marked by (a), (b), (c), (d), (e) and (f).
Response:
- The reason why G3 is presented 3 times because the subfigures showed different aspects of lesions that should be provided in the manuscript
- The difference between the shapes is now marked by (a), (b), (c), (d), (e) and (f).
- Figure 9 is blurry.
Response:
- The figure was enhanced
- Title of Table 7 “Table 7: LCMS/MS data of pirimicarb” should be presented above the table.
Response:
- The title is presented now above the table
- In line 208, “2. Discussion” should be changed to “3. Discussion”
Response:
- Done
- The manuscript suffers from supportive materials and discussion. A comparative study on the previous research findings needs to be included.
Response:
- Done
- In line 348, “Figure 6: experimental design” the number of figure should be corrected.
Response:
- Done
- Although statistical analysis appears to have been performed, there is no text in the methods section providing the details about the tests of the assumptions of ANOVA (normality and homogeneity of variance).
Response:
There is a text in methods section which is “6. Statistical study”, more details were added

Reviewer 2 Report
This manuscript is about the protective effect of Ephedra alata monjauzeana against neurological and reproductive organ damage induced by picimicarbin. I believe the research is well conducted. However, some sections should be improved, as mentioned below.
Some grammatical errors in the manuscript, such as words together, etc. Therefore, it is necessary to review the entire text. It is required to define what EamCE means in the abstract.
Result section
Please place the description of G1..G4 at the bottom of the first figure and in the first table.
Improve the format of the tables (the titles, since each title appears in three lines, table 3) and the same in the data, which appears in two lines.
The captions of each figure should briefly explain each image since only the figures' titles appear. The same happens with the tables; in addition, you must place what means *,**, etc., in each table and figure because the statistical analysis is missing on each graph.
Improve the format of table 6 since it is not easy to identify what each data item (rows) corresponds it.
The graphs must contain information on the ordinate (Y) axis.
The authors should place the chemical formula correctly in table 7.
The data on the dose of pirimicarb is required and not only (1/10 LD...)
Place in each chromatogram the compound to which it corresponds (Fig 9.). At the bottom of the figure, briefly describe what the authors find.
Table 6. What is the purpose of placing protein determination in each group?
The data obtained from the G3 group shows it marked oxidative damage and a reduction in its antioxidant activity; these enzymes also reduce the possibility of trapping reactive oxygen species. I suggest relating the oxidative and antioxidant parameters to ROS production induced by pirimicarb.
Methodology section
It is required to place the concentration of the compounds used and to describe better the methodologies of MDA, SOD, CAT, and GSH.
I consider that when there is no significant difference in the groups, it is unnecessary to place any symbol.
G3 (EanCE+pyrimicarb), were the two compounds given simultaneously? What vehicle contained each of the compounds?
Indicate which is the positive and negative control group in each determination.
Discussion section
Mention the deficiencies that the study has.
Why is GSH in the G2 group increased, Table 6?
It is necessary to discuss with other research where the reproductive toxicology activity was studied.
What is the mechanism by which Ephedra alata monjauzeana protects against the insecticide?
It is unclear if the research aims to study the toxicological effect of the insecticide or the protective effect against the toxicological damage of the insecticide; for this reason, it is essential to reinforce the aim of the research.
Author Response
Dear referees;
We are grateful for your valuable time and useful contribution. We do appreciate the inputs that you have given and we believe that it will definitely help to improve our manuscript. You would find in the paper all the required changes/ additions highlighted in green. Also, you would find below responses for each of your requirements.
This manuscript is about the protective effect of Ephedra alata monjauzeana against neurological and reproductive organ damage induced by picimicarbin. I believe the research is well conducted. However, some sections should be improved, as mentioned below.
Some grammatical errors in the manuscript, such as words together, etc. Therefore, it is necessary to review the entire text. It is required to define what EamCE means in the abstract.
Response:
- Done
Result section
Please place the description of G1..G4 at the bottom of the first figure and in the first table.
Response:
- Done
Improve the format of the tables (the titles, since each title appears in three lines, table 3) and the same in the data, which appears in two lines.
Response:
- The tables were substituted by histograms as recommended the previous reviewer
The captions of each figure should briefly explain each image since only the figures' titles appear. The same happens with the tables; in addition, you must place what means *,**, etc., in each table and figure because the statistical analysis is missing on each graph.
Response:
- Done
Improve the format of table 6 since it is not easy to identify what each data item (rows) corresponds it.
Response:
- Done
The graphs must contain information on the ordinate (Y) axis
Response:
- Done
The authors should place the chemical formula correctly in table 7.
The data on the dose of pirimicarb is required and not only (1/10 LD...)
Response:
- The required detail was added
Place in each chromatogram the compound to which it corresponds (Fig 9.). At the bottom of the figure, briefly describe what the authors find.
Response:
- Done
Table 6. What is the purpose of placing protein determination in each group?
Response:
- The protein determination was mentioned in methods section, it is necessary to give the corresponding result, besides, the values were used to in calculation of oxidative stress parameters.
The data obtained from the G3 group shows it marked oxidative damage and a reduction in its antioxidant activity; these enzymes also reduce the possibility of trapping reactive oxygen species. I suggest relating the oxidative and antioxidant parameters to ROS production induced by pirimicarb.
Response
- An eminent part was added respecting your suggestion.
Methodology section
It is required to place the concentration of the compounds used and to describe better the methodologies of MDA, SOD, CAT, and GSH.
Response
- The required details were added
I consider that when there is no significant difference in the groups, it is unnecessary to place any symbol.
Response
- The symbol was deleted
G3 (EanCE+pyrimicarb), were the two compounds given simultaneously? What vehicle contained each of the compounds?
Indicate which is the positive and negative control group in each determination.
Response
- G4 (EamCE+pyrimicarb), the two products were not given simultaneously:
EamCE (Ephedra alata monjauzeana Crude Extract) was given 1H before giving pirimicarb in order to give it sufficient moment to be absorbed and engender its eventual preventive effect.
The vehicle of each compound was deionized water.
- Both of G1 and G2 are negative controls;
G1: received only deionized water (the negative control of the whole experiment
G2: the negative control of the plant used supposed to give a protective effect, we should to test its effect alone to confirm its safety.
- Actually, there is no positive control, because we’re are testing a molecule for the first time, we were not expecting surly a specific trouble that has a specific treatment to use it as a positive control.
Discussion section
- Mention the deficiencies that the study has.
Response
- Done (in perspectives).
- Why is GSH in the G2 group increased, Table 6?
Response
- The antioxidant potential of the EamCE in G2 has increased the level of GSH
- We have added a new part of discussion that deals with this effect
- It is necessary to discuss with other research where the reproductive toxicology activity was studied.
Response
- Already done, but with other type of pesticides, no human or rat reproductive studies were carried before.
What is the mechanism by which Ephedra alata monjauzeana protects against the insecticide?
Response
- Polyphenols, flavonoids, saponnins and alkaloids (ephedrine ones) from Ephedra alata have many virtues and engender many effects namely, antioxidant, anti-inflammatory, neuroprotective, fertilizing , mood enhancing and many others are sufficiently able to protect tissues and physiologic functions from pirimicarb induced injuries.
It is unclear if the research aims to study the toxicological effect of the insecticide or the protective effect against the toxicological damage of the insecticide; for this reason, it is essential to reinforce the aim of the research.
Response
- Indeed, the current research aimed to evaluate the noxious effects of pirmicarb and eventually the protective effect of Ephedra alata We intended to use it as a protective mean to be a continuity to our previous study that have revealed many biological activities and a richness in polyphenol and flavonoid compounds. The aim and the reason are provided now in the introduction section to reinforce our choice.

Reviewer 3 Report
The manuscript by Khattabi et al., presents novel research on pirimicarb induced behavioral disorders and reproductive toxicities. The study examined both behavioral actions of live rats as well as biochemical and physicochemical parameters. However, I have some concerns regarding the study presented here. The title itself is wrong!! It is not picimicarb but pirimicarb.
1. What is the LD value of pirimicarb? The authors gave a 1/10 LD50 to the rats, but what is the LD50 of pirimicarb for rats?
2. Although the authors examined the preventive effects of Ephedra alata monjauzeana, no rationale or background has been provided as to why the authors specifically chose this plant species to study.
3. What is EamCE? How is it administered in oral gavage? What does 200mg EamCE mean? Was it pellets? Or extracts? Provide more information about this as this is one of the treatment conditions.
4. Why haven’t the authors examined female rats? It would be a comprehensive finding if the study was designed to incorporate both male and female rats and examine the effects of pirimicarb on both male and female rats.
5. When conducting behavioral studies, were all the rats in the group studied/observed individually or were the six rats in a group studied collectively?
6. The graphs presented here have some discrepancies with the table. The errors bars on the bars do not match the values represented in the tables. Please check the data and the figures.
7. How does EamCE protect the cells. The H&E photomicrogrpahs of the G3 and G4 show protective action of EamCE. But the authors have not discussed anything about this. Also, why does the authos say EamCE show intense spermatogenesis, while it appears similar to the control group G1.
8. In the discussion in Lns 312-313, the authors incorrectly conclude that noxious effects were probably due to the action of intact of pirimicarb and from the action of its fragments. This is patently wrong. The fragments 72 and 182 are mass spec fragments and may or may not correspond to their metabolites. The authors are required to also examine the tissue extracts for the metabolites of pirimicarb and provide the chemical structure of the fragments 72 and 182.
9. In Figure 9 what do the Panels A, B, C, D represent? Also, except for Spectra B, all other panels have a spectral intensity of over 1x10^6. Why is it low in Panel B? Also, it would be good if the pirimicarb measured can be quantified. That will give a good measure of pirimicarb uptake and metabolism.
Other comments
The authors have to ensure that there is appropriate spacing between the words. There are several instances of this. Listed few of them as there are way too many to list individually.
Ln 23: Insert space b/w reproductive function
Ln 24: Insert space b/w changes via
Ln 25: Insert space b/w its parameters
Ln 28: When using abbreviations (EamCE) for first time, expand it and put it in parenthesis
Ln 39: Rephrase, the sentence reads confusing
Ln 36: carried out
Ln 78: Write scientific name in italics
Author Response
Dear referee;
We are grateful for your valuable time and useful contribution. We do appreciate the inputs that you have given and we believe that it will definitely help to improve our manuscript. You would find in the paper all the required changes/ additions highlighted in Green. Also, you would find below responses for each of your requirements.
What is the LD value of pirimicarb? The authors gave a 1/10 LD50 to the rats, but what is the LD50 of pirimicarb for rats?
Response:
- The LD50 was added to the manuscript
- Although the authors examined the preventive effects of Ephedra alatamonjauzeana, no rationale or background has been provided as to why the authors specifically chose this plant species to study.
Response:
- An eminent part of Ephedra alatamonjauzeana background and rationale raison were added in introduction section.
- What is EamCE? How is it administered in oral gavage? What does 200mg EamCE mean? Was it pellets? Or extracts? Provide more information about this as this is one of the treatment conditions.
Response:
- The EamCE signification was added where first mentioned, indeed it is a crude extract, it was mentioned that all treatments were given by oral gavage (force-feeding). 200mg of EamCE was dissolved in deionized water.
- Why haven’t the authors examined female rats? It would be a comprehensive finding if the study was designed to incorporate both male and female rats and examine the effects of pirimicarb on both male and female rats.
Response:
- We have not examined female rats as a first step, because of hormonal disturbances including menstrual cycle hormonal interactions in female rats.
- The second reason, in Algeria, 90% of agriculture workers are men, so, it is preferable and more interesting to study the impact of pesticide exposure first in male rats.
- In our subsequent studies we intent to carry a comparative survey between both sexes.
- When conducting behavioral studies, were all the rats in the group studied/observed individually or were the six rats in a group studied collectively?
Response:
- When we were conducting behavioral studies, all the rats in the group were studied/observed individually. Actually, it is impossible and not correct to study them collectively.
- The graphs presented here have some discrepancies with the table. The errors bars on the bars do not match the values represented in the tables. Please check the data and the figures.
Response:
- Done, they were checked
- 7. How does EamCE protect the cells. The H&E photomicrogrphs of the G3 and G4 show protective action of EamCE. But the authors have not discussed anything about this. Also, why does the authors say EamCE show intense spermatogenesis, while it appears similar to the control group G1.
Response:
- We have discussed and added new parts of discussion that elucidate the action of EamCE to exercise a protective effect. Indeed, the composition of this plant rich in polyphenols , flavonoids , saponnins and alkaloids (ephedrine ones) that have many biological activities such as: antioxidant, anti-inflammatory, neuroprotective, fertilizing , mood enhancing, analgesic and many others are synergic and able to protect tissues ,cells and preserving physiologic functions from pirimicarb induced injuries.
- We have said intense spermatogenesis because the rate of testosterone has the most higher value in G2 fed only with EamCE, furthermore, the photomicrographs showed lumen rich with spermatozoids not typically as in G1, but we have replaced “intense” by important to give adequate description and as you have recommended.
- In the discussion in Lns 312-313, the authors incorrectly conclude that noxious effects were probably due to the action of intact of pirimicarb and from the action of its fragments. This is patently wrong. The fragments 72 and 182 are mass spec fragments and may or may not correspond to their metabolites. The authors are required to also examine the tissue extracts for the metabolites of pirimicarb and provide the chemical structure of the fragments 72 and 182.
Response:
- Indeed, it may or may not be that the noxious effect due to the action of metabolites (we have mentioned probably) these latters also may not be the fragments 72 and 182. We have changed this part of discussion. Respected Reviewer, your remark is excellent; and we promise to complete this survey in our later research. We sincerely appreciate it, sir.
- The chemical structure of the fragments 72 and 182 was provided.
- In Figure 9 what do the Panels A, B, C, D represent? Also, except for Spectra B, all other panels have a spectral intensity of over 1x10^6. Why is it low in Panel B? Also, it would be good if the pirimicarb measured can be quantified. That will give a good measure of pirimicarb uptake and metabolism.
Response:
- What represent panels A, B,C, and D was added below the figure.
- The difference between the panels is the intensity of the peaks due to the concentration of pirimicarb; and since our study purpose is qualitative analysis, the intensity is not necessary. The most important is the presence of pirimicarb.
- Probably in further study will work on the quantification.
Other comments
The authors have to ensure that there is appropriate spacing between the words. There are several instances of this. Listed few of them as there are way too many to list individually.
Response:
Ln 23: Insert space b/w reproductive function
- Done
Ln 24: Insert space b/w changes via
- Done
Ln 25: Insert space b/w its parameters
- Done
Ln 28: When using abbreviations (EamCE) for first time, expand it and put it in parenthesis
- Done
Ln 39: Rephrase, the sentence reads confusing
- Done
Ln 36: carried out
- Done
Ln 78: Write scientific name in italics
- Done

Round 2
Reviewer 1 Report
The authors have incorporated the necessary suggestion raised by the reviewers, and now, it can be accepted for publication.
Author Response
Dear Reviewer 1,
We would like to express our full thanks to you for your favorable response.
Reviewer 2 Report
The authors made most of the previous observations; in this version, I have found other observations that I mention below.
· The graphs do not show the title or the measurement units in the Y-axis graphs. Figures 1-3.
· Figure 9.- In each chromatogram, the compound that each compound represents appears; however, the text is tiny.
· The dose of pirimicarb used should be very clear. It is not clear if the dose that was used is 14.5 mg/Kg
· The abbreviation for time is not correct.
· I suggest an additional figure where the proposed mechanism by which EamCe extract protects against pirimicarb-induced damage is placed.
· The positive and negative controls of all the experiments were not answered correctly. Is there an antidote used in case of pirimicarb poisoning?
· On line 442, FST is mentioned; however, in the text, it appears as FTS.
· line 502 is missing a space before reference 87. Same for reference 88 on line 506.
· The chemical formulas are not correct in section 4.6.6.
· Improve the wording of lines 574-576.
· Check what TCA means on line 404.
· It is important that the format of the graphs be improved.
Author Response
We do appreciate the extra changes that you have required and we believe that it will definitely improve our manuscript. You would find in the paper all the required changes/ additions highlighted in purple. Also, you would find below responses for each of your requirements.
- The graphs do not show the title or the measurement units in the Y-axis graphs. Figures 1-3.
Response:
The measurement units were added, the title exist in the figure title (avoiding receptions), besides, further detail for more reading the figure are provided under the title.
- Figure 9.- In each chromatogram, the compound that each compound represents appears; however, the text is tiny.
Response:
The text was adjusted to be clearer
- The dose of pirimicarb used should be very clear. It is not clear if the dose that was used is 14.5 mg/Kg
Response:
It is mentioned now. We have added that it corresponded to 14.5 mg/Kg
- The abbreviation for time is not correct.
Response:
Corrected
- I suggest an additional figure where the proposed mechanism by which EamCe extract protects against pirimicarb-induced damage is placed.
Response:
Done, we would bring to your information, that we have added a section of IL-1β dosage, in order to response to your request of the general mechanism of EamCE intervention against pirimicarb induced damage. Since, we are studying, a neuroimmune endocrine interaction, we presumed that this part was missing. That is way we have added it to endorse the study and schematizing the general approximate phenomena.
- The positive and negative controls of all the experiments were not answered correctly. Is there an antidote used in case of pirimicarb poisoning?
Response:
- The 1rst negative control is group 1 that received deionized water.
- The 2nd negative control concerns EamCE is group 2, in order to confirm that the extract of plant doesn’t deliver any harmful or deleterious effect.
- The 1rst positive control is group 3 which received the pirimicarb as a probable toxic substance.
- The 2nd positive control concerns EamCE is group 4 in order to test its beneficial effect towards eventual negative effect of pirimicarb.
When it comes to the antidote, in this case, we are testing the poisoning of pirimicarb, and the probable protective effect of EamCE, we are not comparing the EamCE with a reference drug. Because, a priori we were not sure that the used concentration (14,5 mg/kg) will engender toxic effects. In another point of view, there is no specific conventional antidote for pirimicarb poisoning.
- On line 442, FST is mentioned; however, in the text, it appears as FTS.
Response:
It is corrected now
- line 502 is missing a space before reference 87. Same for reference 88 on line 506.
Response:
It is corrected now
- The chemical formulas are not correct in section 4.6.6.
Response:
It is corrected now
- Improve the wording of lines 574-576.
Response:
Done
- Check what TCA means on line 404.
Response:
Done
- It is important that the format of the graphs be improved.
Response:
Done